# Striated myocyte structural integrity: Automated analysis of sarcomeric z-discs

**Tessa Altair Morris**[1,2], **Jasmine Naik**[2,3], **Kirby Sinclair Fibben**[4], **Xiangduo Kong**[5], **Tohru Kiyono**[6], **Kyoko Yokomori**[5], **Anna Grosberg**[1,2,3,4,7]*

**1** Center for Complex Biological Systems, University of California, Irvine, Irvine, California, United States of America, **2** Edwards Lifesciences Center for Advanced Cardiovascular Technology, University of California, Irvine, Irvine, California, United States of America, **3** Department of Chemical and Biomolecular Engineering, University of California, Irvine, Irvine, California, United States of America, **4** Department of Biomedical Engineering, University of California, Irvine, Irvine, California, United States of America, **5** Department of Biological Chemistry, School of Medicine, University of California, Irvine, Irvine, California, United States of America, **6** Division of Carcinogenesis and Cancer Prevention, National Cancer Center Research Institute, Tsukiji, Chuo-ku, Tokyo, Japan, **7** NSF-Simons Center for Multiscale Cell Fate Research, University of California, Irvine, Irvine, California, United States of America

* grosberg@uci.edu

**Data Availability Statement:** The data can be found at https://doi.org/10.7280/D12Q2X, and the code can be found at https://github.com/Cardiovascular-Modeling-Laboratory/zlineDetection.

## Abstract

As sarcomeres produce the force necessary for contraction, assessment of sarcomere order is paramount in evaluation of cardiac and skeletal myocytes. The uniaxial force produced by sarcomeres is ideally perpendicular to their z-lines, which couple parallel myofibrils and give cardiac and skeletal myocytes their distinct striated appearance. Accordingly, sarcomere structure is often evaluated by staining for z-line proteins such as $\alpha$-actinin. However, due to limitations of current analysis methods, which require manual or semi-manual handling of images, the mechanism by which sarcomere and by extension z-line architecture can impact contraction and which characteristics of z-line architecture should be used to assess striated myocytes has not been fully explored. Challenges such as isolating z-lines from regions of off-target staining that occur along immature stress fibers and cell boundaries and choosing metrics to summarize overall z-line architecture have gone largely unaddressed in previous work. While an expert can qualitatively appraise tissues, these challenges leave researchers without robust, repeatable tools to assess z-line architecture across different labs and experiments. Additionally, the criteria used by experts to evaluate sarcomeric architecture have not been well-defined. We address these challenges by providing metrics that summarize different aspects of z-line architecture that correspond to expert tissue quality assessment and demonstrate their efficacy through an examination of engineered tissues and single cells. In doing so, we have elucidated a mechanism by which highly elongated cardiomyocytes become inefficient at producing force. Unlike previous manual or semi-manual methods, characterization of z-line architecture using the metrics discussed and implemented in this work can quantitatively evaluate engineered tissues and contribute to a robust understanding of the development and mechanics of striated muscles.

**Funding:** This work was supported by the Edwards Lifesciences Center for Advanced Cardiovascular Technology's NIH/NHLBI T32HL116270 Training Grant (PI: Hughes, Trainee: TAM), NIH R01 HL129008 2015-2020 (MPI: Grosberg, Zaragoza) (AG), NSF DMS1763272 & Simons Foundation (594598, PI: QNie) (AG), National Institutes of Health (AR067636 and AR071287 to KY), and National Cancer Center Research and Development Fund (23-B-1 and 23-A-38 to TK). The funders had no role in study design, data collection and analysis, decision to publish, or preparation of the manuscript.

**Competing interests:** The authors have declared that no competing interests exist.

## Author summary

Structural evaluation of sarcomeres is fundamental to the study of striated muscle. However, due to limitations of current analysis methods, the mechanisms by which sarcomere order can impact contraction and the characteristics of sarcomere architecture that should be used to assess striated myocytes have not been fully explored. Furthermore, it is not clear what aspects of sarcomere architecture are considered by the experts when qualitatively evaluating striated muscle tissues. Therefore, we developed a computational structural assay in MATLAB, `ZlineDetection`, to evaluate sarcomere architecture by both extracting sarcomeric z-lines from images and providing metrics that encapsulate different aspects of z-line architecture that an expert would evaluate when judging the quality of the tissue. The sarcomere structure of both patient-specific skeletal muscle and rat cardiomyocytes were evaluated with differences among engineered cells and tissues quantified using novel and established metrics. As a result, a mechanism by which highly elongated cardiomyocytes become inefficient at producing force was elucidated. `ZlineDetection` identifies and quantifies the characteristics used by experts for evaluation and thus it will lead to more rigorous differentiation methods and tissue comparison across labs and contribute a robust understanding of how structure affects mechanical function.

## Introduction

Assessment of cellular morphology and structure is fundamental to the study of striated muscles. It has been used to characterize the developmental stage [1–3], engineered tissues [4–8], effects of disease [9–12] or injury [13–16], and treatment with pharmacological agents [17] as well as used to predict function [18–20]. Indeed, the ability of striated muscle cells to contract is dependent on the nearly crystalline order of its cytoskeletal components [21, 22], which makes evaluation of structure paramount. Skeletal and cardiac myocytes are composed of parallel myofibrils, which are spanned by repeating sarcomere units that produce a contractile force parallel to the thick myosin filaments as they slide past the thin actin filaments [23, 24]. Consequently, myofilament disorganization has been shown to have a critical role in contractile impairment [25, 26]. The uniaxial force produced by sarcomeres is ideally perpendicular to their z-lines, which couple parallel myofibrils and give cardiac and skeletal myocytes their distinct striated appearance [27]. Accordingly, sarcomere structure is often evaluated by staining for z-line proteins such as $\alpha$-actinin [1, 5–8, 18–20, 23, 24, 28–35]. A disruption in alignment or registration of z-lines across neighboring myofibrils has been observed in the ventricles of failing hearts [26]. However, the mechanism by which sarcomere and by extension z-line architecture can impact contraction and which characteristics of z-line architecture should be used to assess striated myocytes has not been fully explored.

Certain aspects of z-line architecture quantifiable from flourescently stained images have been used to evaluate engineered tissues and to predict function. In particular, because the uniaxial force of sarcomeres is maximized when they are all oriented in the same direction, the orientational order of the z-line protein $\alpha$-actinin has been used extensively as a metric [3, 6, 18, 30, 36]. Similarly, the correlation between the orientational order of z-lines and actin fibrils is also an important metric because the relative orientation of these structures transitions from parallel to perpendicular during development [5, 8, 37, 38]. The relative location and lateral alignment of z-lines in neighboring myofibrils is also hypothesized to influence contractile function and to be an important metric to assess myofibril formation [20, 32–34, 39–42]. In

particular, theoretical models have been established to explain why z-lines of neighboring myofibrils tend to register during development [40] and the impact of z-line registration on contractile function [20, 39]. While theoretical models and experiments have provided insight into which facets of z-line architecture are important to characterize striated myocytes, research has been hindered due to limitations of current analysis methods to quantify z-line architecture, which require manual or semi-manual handling of images [3, 5, 6, 8, 18–20, 24, 30, 32–34, 41].

Quantification of z-line architecture involves accurately extracting z-lines from an image. Although researchers experienced with striated myocytes have the skill to manually trace and extract z-lines, it is a low throughput method and highly susceptible to variability between observers. Previous work has utilized techniques from the field of image analysis such as topology-preserving thinning and edge detection to extract features, such as z-lines, from an image [43–45]. However, these methods have not been optimized for striated myocytes because they include regions of off-target staining, which would be ignored by an expert tracing the z-lines. Once the z-lines have been automatically or manually identified in an image, there is an additional challenge of choosing metrics to summarize overall z-line architecture [3, 5]. Although experts can qualitatively score tissue quality, it is ambiguous which aspects of sarcomere architecture are being considered and why those elements are important. Auxiliary hurdles include biological variability, as well as variability in staining and image quality. These challenges have gone largely unaddressed in previous work, which leave researchers without robust, repeatable tools to assess z-line architecture.

In this work, we investigated which aspects of sarcomere architecture experts use to evaluate the quality of striated tissue, including those implicated in existing theoretical models of muscle fiber contraction. To do this, we developed ZlineDetection in MATLAB, the first fully automatic computational protocol to both isolate z-lines and characterize z-line architecture. Isolating z-lines involved constructing a biologically motivated approach to segment (i.e. remove) off-target staining without the need for user input. Along with reporting existing metrics such as z-line orientational order, ZlineDetection was used to calculate the fraction of $\alpha$-actinin staining that composes well-formed z-lines and find the location and length of z-lines of neighboring myofibrils that are both registered and continuous. Additionally, analysis of isolated cardiomyocytes with variable aspect ratios were compared with published results [32]. Finally, ZlineDetection was used to differentiate among tissues engineered to be anisotropically or isotropically organized, but well-formed, and those engineered to be malformed. By building on previous image analysis methods and establishing new metrics, ZlineDetection automatically and quantitatively assesses sarcomere architecture, and can be used by researchers imaging z-lines with fluorescent staining.

## Materials and methods

### Ethics statement

All animals were treated according to the Institutional Animal Care and Use Committee of UCI guidelines (IACUC Protocol No. 2013-3093). It also followed recommendations of the NIH Guide for the Care and Use of Laboratory Animals and was in accordance with existing federal (9 CFR Parts 1, 2, & 3), state, and city laws and regulations governing the use of animals in research and teaching. The adult Sprague-Dawley rat was euthanized by $CO_2$ inhalation followed by cervical dislocation at a ULAR facility. Dam's euthanasia was done prior to pup sacrifice in order to minimize the stress the dams experience when their pups are taken. The rat pups were then immediately taken to our core lab where each 2 day old neonatal rat pup was euthanized by decapitation. This euthanasia method adheres to the current most humane

standards, which maintain scientific validity of the cell cultures as stated in the "AVMA Guidelines for the Euthanasia of Animals: 2013 Edition" (published by the American Veterinary Medical Association).

## Substrate preparation and extracellular matrix patterning

Substrates were fabricated for structural studies as described previously [5, 8, 19, 24, 46]. Briefly, large cover glass (Brain Research Laboratories, Newton, MA) was cleaned by sonicating, then spin coated with 10:1 Polydimethylsiloxane (PDMS; Ellsworth Adhesives, Germantown, WI). The PDMS coated cover glass was then placed in a 60˚C oven to cure overnight (12 h). The cover glass was then cut into smaller individual coverslips to fit in a 12 well plate. Fibronectin (FN; Fischer Scientific Company, Hanover Park, IL) was patterned onto the coverslips in lines 20 μm wide with 5 μm gaps or islands of various aspect ratios using microcontact printing [47]. The PDMS stamps were then sonicated in ethanol and coated with 0.1 mg/mL drops of FN. After being incubated for 1 h and dried using compressed nitrogen, FN was printed onto the PDMS coated coverslips that were previously exposed to UV light (Jelight Company, Irvine, CA) for 8 min. Finally, the stamped coverslips were submerged in a solution of 5 g Pluronics F-127 (Sigma Aldrich, St. Louis, MO) dissolved in 0.5 L sterile water for 5 min and then rinsed three times with room temperature phosphate-buffered saline (PBS; Life Technologies, Carlsbad, CA). Isotropic tissue samples were made by coating coverslips with a uniform layer of FN [5].

## Cardiomyocyte culture

Ventricular myocardium was extracted from two day old neonatal Sprague Dawly rats (Charles River Laboratories Wilmington, MA) under sterile conditions and in accordance with the guidelines of the Institutional Animal Care and Use Committee of University of California, Irvine (Protocol No. 2013-3093). Cardiomyocytes were then isolated from the ventricular myocardium as described previously [5, 8, 19, 24]. Briefly, after rinsing the ventricular tissue in Hanks' balanced salt solution buffer (HBSS; Life Technologies, Carlsbad, CA), the tissues were incubated overnight (12 h) at 4˚C in a 1 mg/mL trypsin solution (Sigma Aldrich, Inc., Saint Louis, MO) dissolved in HBSS. After neutralizing the trypsin in warmed 10% fetal bovine serum (FBS; ThermoFisher, Grand Island, NY) M199 culture media (Invitrogen, Carlsbad, CA), the tissue was washed four times with 1 mg/mL collagenase (Worthington Biochemical Corporation, Lakewood, NJ) dissolved in HBSS. Isolated cells were centrifuged at 1200 rpm for 10 min and re-suspended in chilled HBSS, before being centrifuged again at 1200 rpm for 10 min. The cells were then re-suspended in warm 10% FBS M199 culture media and purified through three consecutive preplates. Cells were then counted and seeded onto FN coated coverslips. The seeding density used to produce confluent monolayers was 1400 cells/mm$^2$, the density for sparse tissue was 350 cells/mm$^2$, and the density for isolated cardiomyocytes was 200 cells/mm$^2$. At 24 h post-seeding, dead cells were washed away with PBS and the remaining cells were incubated in 10% FBS M199 media. After 24 h, the 10% FBS media was replaced with 2% FBS M199 media. As described previously [28], for cells treated with 2,3-butanedione 2-monoxime (BDM; Sigma Aldrich, Inc., St. Louis, MO), the culture media containing 10 mM BDM was prepared by adding 1/500 volume 5 mol/L BDM dissolved in dimethylsulfoxide (DMSO; Sigma Aldrich, Inc., St. Louis, MO) and stored at -20˚C. BDM was left to interact with cells for two days before fixing.

## Skeletal muscle preparation

Human primary cultured healthy control myoblasts were immortalized using hTERT with p16$^{INK4a}$-resistant R24C mutant CDK4 and cyclin D1 as previously described [48]. After

immortalization, CD56-positive cells were selected by magnetic-activated cell sorting conjugated with anti-CD56 antibody (130-050-401, Miltenyi Biotec). Myoblast differentiation was induced as previously described [49]. Briefly, CD56-positive cells were plated onto coverslips at a seeding density of ~2.5 x $10^5$ cells/mL in 2 mL of growth medium (high glucose DMEM (11965, Gibco) supplemented with 20% FBS (FB-02, Omega Scientific, Inc.), 1% Pen-Strep (15140122, Gibco) and 2% Ultrasor G (67042, Crescent Chemical Co.)) in each well of a 12-well dish. Approximately 12-16 h later, differentiation was induced using high glucose DMEM medium supplemented with 2% FBS and ITS supplement (insulin 0.1%, 0.000067% sodium selenite, 0.055% transferrin, 51300044 Invitrogen). Fresh differentiation media was changed every day.

### Fixing, immunostaining, and imaging

After 72 h (cardiomyocytes) or 14 days (skeletal muscle) in culture, cells were fixed in warm 4% paraformaldehyde (Fisher Scientific, Hanover Park, IL) supplemented with 0.001% Triton X-100 (Sigma-Aldrich, Inc., St. Louis, MO) in PBS for 10 min. Cells were rinsed three times in room temperature PBS for 5 min and then stained for actin (Alex Fluor 488 Phalloidin; Life Technologies, Carlsbad, CA), sarcomeric $\alpha$-actinin (Mouse Monoclonal Anti –actinin; Sigma Aldrich, Inc., St. Louis, MO), nuclei (4',6'-diaminodino-2-phenlyinodol (DAPI; Life Technologies, Carlsbad, CA), and FN (polyclonal rabbit anti-human fibronectin; Sigma Aldrich, Inc., St. Louis, MO). Secondary staining was applied using tetramethylrhodamine-conjugated goat anti-mouse IgG antibodies (Alexa Fluor 633 Goat anti-mouse or Alexa Fluor 750 Goat anti-mouse; Life Technologies, Carlsbad, CA) and goat anti-rabbit IgG antibodies (Alexa Fluor 750 goat anti-rabbit or Alexa Fluor 633 Goat anti-rabbit; Life Technologies, Carlsbad, CA) for a 1-2 h incubation. The coverslips containing the immunostained cells were then mounted onto a microscope slide preserved with prolong gold antifade reagent (Life Technologies, Carlsbad, CA). The images were collected using an IX-83 inverted motorized microscope (Olympus America, Center Valley, PA) with an UPLFLN 40x oil immersion objective (Olympus America, Center Valley, PA) and a digital CCD camera ORCA-R2 C10600-10B (Hamamatsu Photonics, Shizuoka Prefecture, Japan). The resolution of the images taken with the 40x oil objective was ~6 pixels/μm. Ten to fifteen fields of view were randomly acquired for every sample. Raw images and other data have been deposited in the Dryad repository: https://doi.org/10.7280/D12Q2X [50].

### Statistical analysis

To determine statistical significance, one-way analysis of variance (ANOVA) with Tukey's Test was performed in R version 3.5.2 using RStudio Version 1.1.463. A p-value less than 0.05 was considered significant. Sample size calculations were also performed using R with a power of 0.95.

### Image processing

We developed the following analysis procedure, we named ZlineDetection, which we made available on Github (https://github.com/Cardiovascular-Modeling-Laboratory/zlineDetection). ZlineDetection was implemented in MATLAB version 9.5.0.1033004 (R2018b) (MathWorks, Natick, MA). Parameters are listed and described in S1 Table and the user guide can be found in the Github repository for ZlineDetection.

**Extraction of $\alpha$-actinin skeleton and orientation.** The z-line architecture was analyzed after extracting the binary skeleton and orientation vectors of the z-lines in images of $\alpha$-actinin stained cardiac tissue. The process of extracting the $\alpha$-actinin binary skeleton (S1 Fig) was

adapted from an image analysis protocol established for fibrillar materials in MATLAB [51]. Briefly, gray-scale images were smoothed using coherence enhancing anisotropic diffusion filtering, which calculates the eigenvectors of the image Hessian to direct diffusion and uses a finite difference scheme to perform the diffusion, repeating until diffusion time was reached [52–54]. The diffusion time and smoothing parameters were selected by choosing parameters that resulted in skeletons with the highest similarity [55] to manually traced sections of cardiac tissue (S2 Fig). The parameters selected minimized the following equation:

$$E_p = \frac{1}{n} \sum_i^n (S_{i,p} - S_{i,\max})^2.$$  (1)

In Eq 1, the error ($E_p$) for parameter set $p$ is the squared difference between the similarity of image $i$ for the current parameter set ($S_{i,p}$) and the maximum similarity for image $i$ ($S_{i,\max}$), divided by the number of images ($n$). After contrast enhancement using top hat filtering [56], the background was removed using a surface interpolation [57] and then binarized by adaptive thresholding. Finally, the binarized images were thinned to one pixel width and trimmed in order to obtain the $\alpha$-actinin skeleton. Local orientation was estimated from the diffusion filtered image using a least mean square orientation estimation algorithm [58, 59].

**Actin orientation detection.**   The orientation of actin at each pixel was calculated as described previously [18, 30]. Briefly, images were filtered with a Gaussian kernel and then normalized to have zero mean and unit standard deviation [59]. The orientation was then estimated using a least mean square orientation estimation algorithm [58, 59].

**Actin guided segmentation.**   In order to obtain the z-line skeleton, off-target $\alpha$-actinin staining was segmented by using the local orientation of actin. Local orientation of actin was computed by breaking images into grids and computing the structural tensor for the orientation vectors $\vec{r}(x, y)$ in each grid. Local orientation was defined as the director in each grid, which is the eigenvector corresponding to the maximum eigenvalue of tensor $\mathbb{T}$:

$$\mathbb{T} = \left\langle 2 \begin{bmatrix} r_{i,x} r_{i,x} & r_{i,x} r_{i,y} \\ r_{i,x} r_{i,y} & r_{i,y} r_{i,y} \end{bmatrix} - \begin{bmatrix} 1 & 0 \\ 0 & 1 \end{bmatrix} \right\rangle.$$  (2)

The $\alpha$-actinin orientation vectors were compared to the local orientation of actin by taking the dot product:

$$\gamma = \vec{p}_{\text{actin}} \cdot \vec{q}_{\alpha-\text{actinin}}.$$  (3)

An $\alpha$-actinin pixel was considered off-target staining when $\gamma$ in Eq 3 was greater than a threshold, indicating $\vec{p}_{\text{actin}}$ and $\vec{q}_{\alpha-\text{actinin}}$ were too close to parallel.

**Orientational order parameter.**   As described previously [24, 60], the orientational order of constructs in an image was quantified by the maximum eigenvalue of the structural tensor $\mathbb{T}$ (Eq 2), termed the Orientational Order Parameter (OOP).

## Continuous z-line detection

The continuous z-lines were detected in images of $\alpha$-actinin stained cardiac tissue after generation of the z-line binary skeleton and corresponding orientation vectors. We created and implemented the following method in MATLAB to group z-line orientation vectors based on their location and direction. Briefly, each orientation vector ($\vec{v}_i$), with the exception of those located at the edges of the image, have eight neighboring pixels that surround it. Six "candidate neighbors" were chosen by excluding the neighboring pixels that were positioned in the

directions perpendicular to the angle ($\pm\theta_0$) corresponding to $\vec{v}_i$. These "candidate neighbors" were then narrowed down to two by selecting one neighbor in each direction that had the highest dot product value, meaning that the orientation vector at that position is the most parallel to $\vec{v}_i$. These sets of orientation vectors and their neighbors were then iteratively grouped into continuous z-lines.

### Nuclei counting

The nuclei per area was computed by summing the number of nuclei across all fields of view of a coverslip and then dividing by the total area in square millimeters.

## Results

### Automating isolation of z-lines

Accurately extracting z-lines from images of striated myocytes, including primary cardiomyocytes (Fig 1Ai) and patient-specific skeletal muscles (Fig 1Aii), is essential to evaluating their z-

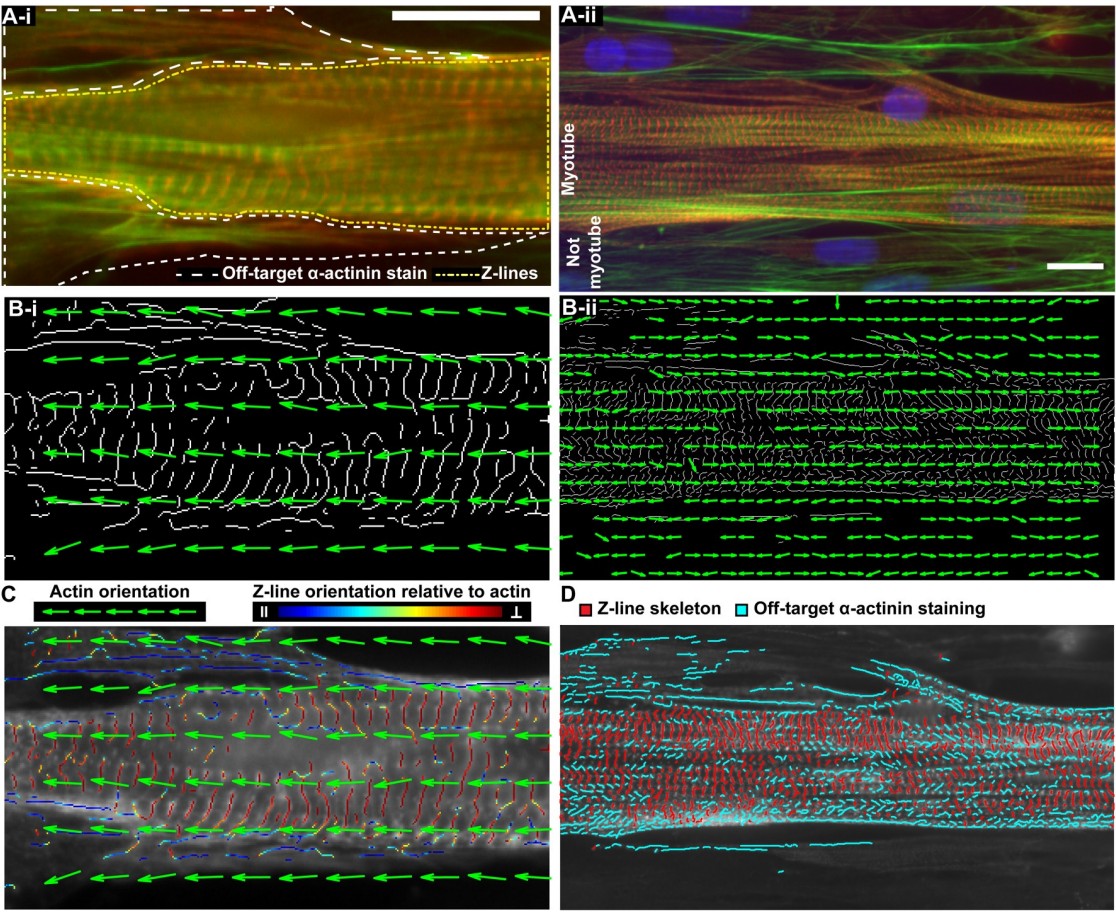

**Fig 1. Actin orientation guided segmentation of the α-actinin skeleton. A**, Images of cardiac **(Ai)** and skeletal muscle **(Aii)** stained for actin fibrils (green), α-actinin (red), and nuclei (blue). In **Ai**, off-target α-actinin stain is outlined in white and the region containing z-lines is outlined with a yellow dashed-dotted line. **B**, The orientation of the actin fibrils in **(A)**, represented by green arrows, plotted on top of the α-actinin skeleton. **C**, Actin orientation vectors (green) overlaid on α-actinin stained cardiac tissue **(Ai)** where each pixel in the α-actinin binary skeleton **(Bi)** is colored according to its orientation relative to local actin orientation from parallel (dark blue) to perpendicular (red) as indicated by the colorbar. **D**, Skeletal muscle shown in **Aii** with off-target α-actinin staining (blue) and z-lines (red). Scale bars: 15 μm.

line architecture. In images of striated muscle cells stained for $\alpha$-actinin, a protein within the sarcomeric z-lines (Fig 1A), $\alpha$-actinin appears as striations approximately perpendicular to actin fibrils (Fig 1A, *green*). However, in addition to the sarcomeric z-lines (Fig 1Ai, *red vertical striations in region outlined in yellow*), $\alpha$-actinin tends to be present at the cell boundaries and along immature stress fibers that do not have fully developed z-lines [2] (Fig 1Ai, *white outline*). In contrast to z-lines, regions of off-target $\alpha$-actinin staining are often oriented along the direction of actin (Fig 1A, *green*), rather than perpendicular [5, 37]. Consequently, when $\alpha$-actinin images are condensed into their binary skeletons [51, 58, 59] (Fig 1B), off-target $\alpha$-actinin staining appears as long line segments oriented perpendicular to the z-lines and parallel to the orientation of actin fibrils, represented by green arrows (Fig 1B). Including regions of off-target $\alpha$-actinin staining in quantitative evaluation of z-line architecture, in particular assessment of orientational order, causes results to be less accurate.

While image processing procedures such as anisotropic diffusion filtering provided a solution to remove imaging noise [51, 52], it was not capable of excluding off-target $\alpha$-actinin staining. However, it is possible to classify $\alpha$-actinin positive pixels as either off-target staining or z-lines according to their orientation relative to their local actin fibrils. Therefore, the orientation of each pixel in the $\alpha$-actinin skeleton was compared to the local orientation of actin (Eq 3), where local actin orientation was defined as the director in a region that fits the z-lines of two sarcomeres in cardiomyocytes (~5μm x 5μm). Using this method, pixels in the z-line skeleton that had an orientation more parallel to the local direction of actin ($\gamma \geq 0.7$, Eq 3), were classified as off-target $\alpha$-actinin staining and eliminated from the z-line binary skeleton (Fig 1C, *dark blue*), and pixels with an orientation more perpendicular to actin were classified as z-lines (Fig 1C, *red*). The orientation of actin fibrils can then be used to remove off-target staining from the $\alpha$-actinin binary skeleton (Fig 1D).

## Metrics to quantify z-line architecture

With the z-lines automatically isolated from $\alpha$-actinin stained images, it was possible to explore how to summarize other facets of z-line architecture. The sarcomere orientational order parameter (OOP) has been used to indirectly evaluate z-line architecture [3, 6, 8, 18, 30, 36]. However, because the sarcomere OOP included regions of off-target staining, the OOP would be lower in tissues with more off-target staining regardless of the z-line organization. Therefore, the OOP of the isolated z-lines more accurately captures the sarcomere organization. Mathematically, the OOP is calculated the same way regardless of the construct (Eq 2), thus the OOP will better represent z-line orientational order when it is calculated from only the z-lines (Fig 1C and 1D).

In addition to more accurately extracting z-lines, actin guided segmentation provides a metric for how much off-target staining is present in images of striated muscles. The amount of the original $\alpha$-actinin skeleton that remains after actin orientation guided segmentation (Fig 1D, *red*), or the *z-line fraction* quantifies the amount of off-target staining in an image:

$$\text{z-line fraction} = \frac{N_z}{N_\alpha}. \tag{4}$$

In Eq 4, $N_\alpha$ is the number of pixels in the $\alpha$-actinin skeleton, and $N_z$ is the number of pixels in the $\alpha$-actinin skeleton after actin orientation guided segmentation, also referred to as the z-line skeleton. When every pixel in the $\alpha$-actinin skeleton is approximately perpendicular to its local actin fibrils, the z-line fraction will be 1, and if no pixel in the skeleton is approximately perpendicular, the z-line fraction will be 0.

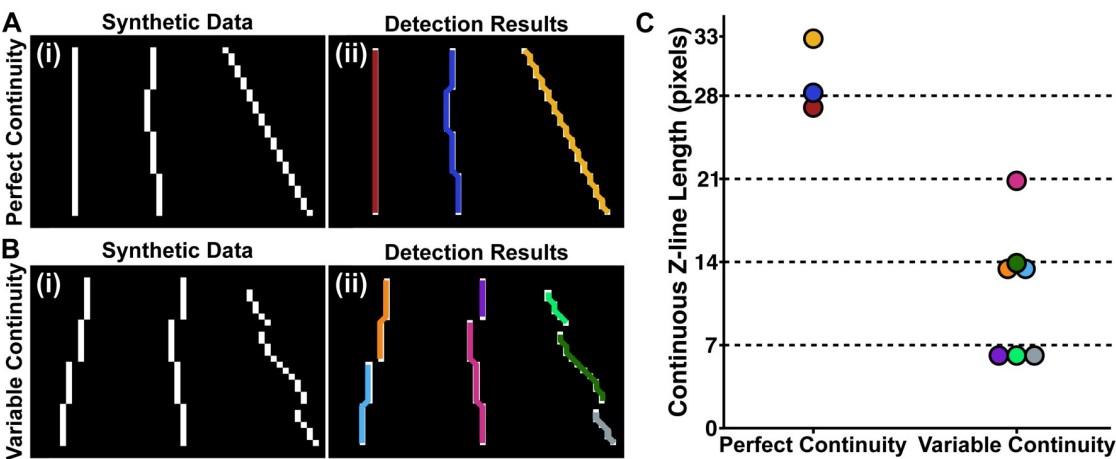

**Fig 2. Automatic detection of continuous z-line lengths.** Perfectly (**A**) and variably (**B**) continuous synthetic data. **Ai** and **Bi**, Synthetic data composed of four registered and continuous segments that were each seven pixels long. Segments that were shifted over by one pixel were treated as continuous. **Aii** and **Bii**, Continuous lines plotted as distinct colors on top of the synthetic data. **C**, The length of each continuous line detected in **Aii** and **Bii**, with the colors corresponding to those in **Aii** and **Bii**. The dashed lines indicate the number of pixels that composed continuous segments.

Another metric of interest was the distribution of continuous z-line lengths. We developed an algorithm to detect and measure the lengths of continuous z-lines, which was validated using synthetic data created to simulate perfectly (Fig 2A) and variably continuous z-lines (Fig 2B). Each segment consisted of 7 pixels, where a shift of one pixel between segments was considered continuous (Fig 2Ai) and a shift of two or more was not considered continuous (Fig 2Bi). As z-lines are not always oriented perpendicular to the image, the synthetic data included segments with variable orientations (Fig 2Ai and 2Bi, *right*). Continuous lines were measured (in pixels) and represented by distinct colors plotted on top of the synthetic data (Fig 2Aii and 2Bii), with the corresponding lengths color-coded in Fig 2C. Measurement of the continuous lines was sensitive to rotation and variation between segments, as indicated by the difference in lengths (Fig 2C) between the red, dark blue, and yellow lines in Fig 2Aii. The automated protocol accurately reported the position and length of synthetic data that was generated to mimic the appearance of perfectly and non-continuous z-lines in images of $\alpha$-actinin stained cardiac tissues (Fig 2).

## Evaluation of single cells with variable aspect ratios

In evaluating the automated method to isolate z-lines from $\alpha$-actinin stains and measuring continuous z-lines, it is useful to understand how the results relate to other aspects of z-line architecture that can be quantified. For example, in investigating the relationship between maximal traction force and cardiomyocyte aspect ratio, Kuo et al. measured the median z-line registration length in order to summarize lateral registration [32]. Cellular aspect ratio is tightly regulated (~7:1) in healthy ventricles [61–63], but increases [62, 63] or decreases [64, 65] in some types of heart disease. While related to continuous z-line length, the z-line registration length will invariably be longer, as z-lines that are not continuous can be registered. However, because Kuo et al. found that both lateral registration of z-lines and maximal traction force varied with cellular aspect ratio, it was interesting to evaluate if aspect ratio impacts other facets of z-line architecture accessible with ZlineDetection. Thus, the analysis was completed for cardiomyocytes with constant area (2500 μm²) and variable aspect ratios (Fig 3A), which were created previously [5]. Because there was variability in the spread of the cells and

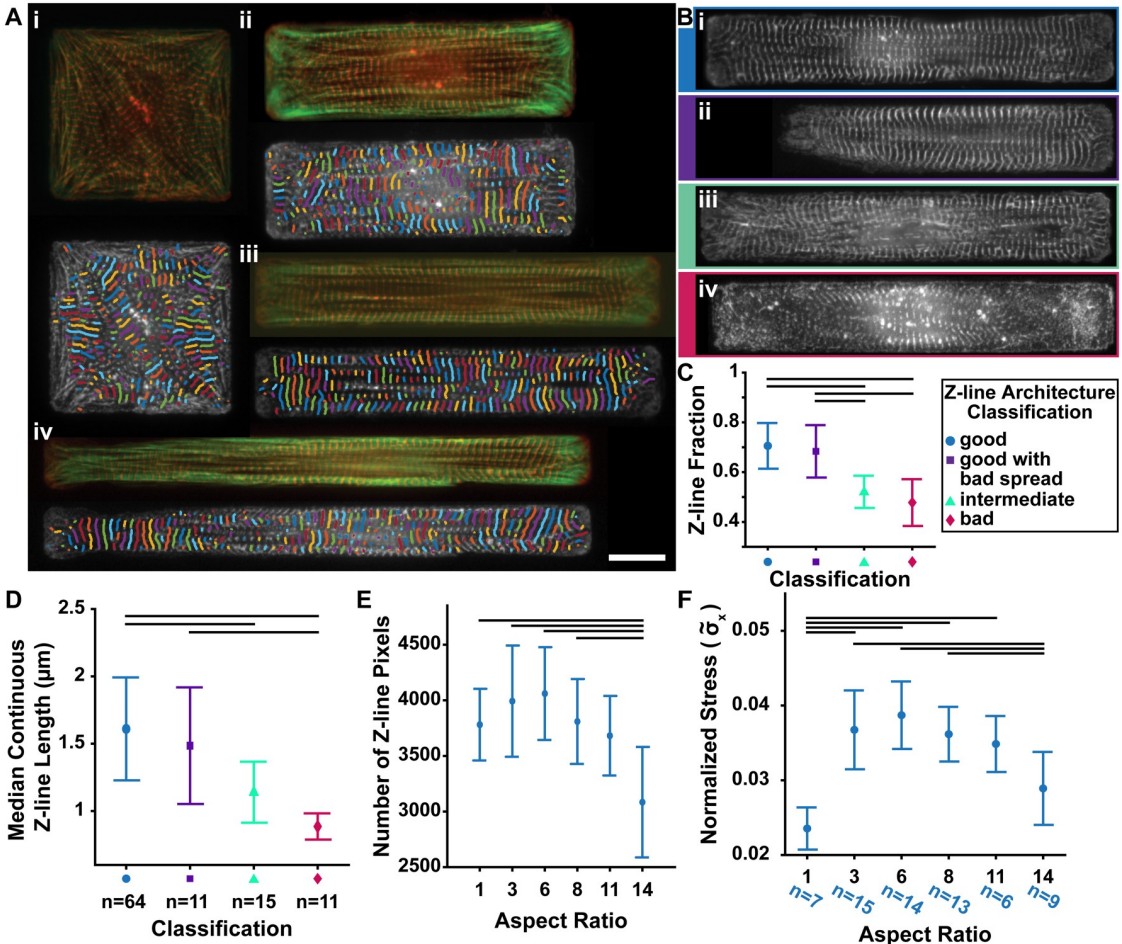

**Fig 3. Analysis of cardiomyocytes with variable aspect ratios. A**, **(top)** Images of single cells with area 2500 μm², but variable aspect ratios (**(Ai)** 1:1, **(Aii)** 3:1 **(Aiii)** 6:1 **(Aiv)** 11:1) stained for actin (green) and α-actinin (red) and their corresponding continuous z-lines (**bottom**). **B**, Representative α-actinin stained cardiomyocytes (extracellular matrix (ECM) island ~6:1 aspect ratio) for each z-line architecture classification: **(Bi)** good z-line architecture, **(Bii)** good z-line architecture with bad spread in ECM island, **(Biii)** intermediate z-line architecture, and **(Biv)** immature, underdeveloped (bad) z-line architecture. **C**, Average z-line fraction for cells. **D**, Average median continuous z-line length for cells (n = 101) within each classification of z-line architecture as described in **C**. **E**, Mean and standard deviation of the number of z-line pixels for the good cells of each aspect ratio. **F**, Mean and standard deviation of the estimated force for the good cells of each aspect ratio. Groups were compared using ANOVA with Tukey's test p <0.05 (black bars in **C**, **E**, and **F**). Scale bar: 15 μm.

the maturity of myofibrils within cardiomyocytes of the same area and aspect ratio, prior to running the analysis, we manually classified cells as having good z-line architecture (Fig 3A and 3Bi), good z-line architecture with bad spread (Fig 3Bii), intermediate z-line architecture (Fig 3Biii), or immature, underdeveloped (bad) z-line architecture (Fig 3Biv). In manual analysis, cells that are not well spread or fully mature are usually eliminated, however our automated analysis made it possible to analyze these cells in addition to cells with good z-line architecture and cell spread. Indeed, the z-line fraction was significantly different between cells with z-line architecture that was good and those classified as intermediate or bad (Fig 3C), indicating, unsurprisingly, cells with good z-line architecture contain less off-target α-actinin staining. Thus, the z-line fraction can be used to filter out single cells with unsatisfactory sarcomeric architecture.

For each of the classifications, cardiomyocytes exhibited a skewed distribution of continuous z-line lengths, with shorter continuous z-lines dominating the distribution (S3 Fig), similar to the distribution of z-line registration lengths reported by Kuo et al. [32]. Therefore, the median continuous z-line length was selected as the metric to compare conditions, which captured the differences between cells with good z-line architecture and cells with intermediate or bad z-line architecture (Fig 3D). When comparing continuous z-line lengths between cells of different aspect ratios, as expected, the median continuous z-line length was lower than the median z-line registration length reported by Kuo et al. (S3 Fig). ZlineDetection can also output the total z-line pixels identified, which can be especially useful when comparing cells of the same area (Fig 3E).

In order to compare our results with the maximal traction forces measured by Kuo et al., we estimated the expected stress generated along the major axis of a cell based on sarcomere architecture [19]. Each z-line orientation vector was represented by its angle ($\theta_i$). Because sarcomeres produce a force approximately perpendicular to their z-lines, we assumed that at each z-line pixel, the force produced by a sarcomere was proportional to the vector perpendicular ($\theta_i + \frac{\pi}{2}$) to the z-line orientation pseudo-vector ($\theta_i$). Thus the stress generated along the major axis of cells with the same area was proportional to the sum of the x-component of the force produced by a sarcomere at each z-line pixel divided by the total number of pixels in the cell area (Eq 5):

$$T_x \propto \tilde{\sigma}_x = \frac{\sum_{i=1}^{N_z} \sqrt{\left(\cos\left(\theta_i + \frac{\pi}{2}\right)\right)^2}}{N_T}.$$

(5)

In Eq 5, the normalized stress, $\tilde{\sigma}_x$, is proportional to the maximal traction force ($T_x$) along the major axis of the cell, $N_z$ is the number of z-line pixels, $N_T$ is the total number of pixels in the cell area, and $\theta_i$ is the orientation of the $i^{\text{th}}$ z-line orientation vector. Consistent with published results [32], the theoretical model captures the lower stresses produced by square cells (1:1), which have low z-line orientational order (Fig 3Ai and S3D Fig). Additionally, consonant with experimental measurements, highly elongated cells (14:1) were predicted to produce a weaker force (Fig 3F). Although every cell was engineered to have the same area, highly elongated cells contained fewer z-lines (Fig 3E), indicating disrupted myofibril formation at this aspect ratio. However, this force estimate does not capture the strong peak stress that occurs in cells with an aspect ratio of ~7:1 (6:1 or 8:1), which could be due to the large biological variability between cells within each aspect ratio. However, it is also likely the assumption that each z-line pixel is independent, meaning that $\tilde{\sigma}_x$ does not account for either continuity or registration, could account for the lack of a maximum at the ~7:1 aspect ratio.

## Z-line architecture in engineered tissues

A motivation for developing this automated analysis protocol was to quantify and compare z-line architecture in tissues using unbiased, quantitative metrics. Previously, characterizing z-line architecture in tissues required manual removal of off-target staining [3, 5, 8, 18, 30, 32], however, with ZlineDetection it is now possible to quantify changes to z-line architecture without the need for user input. The contractile force of engineered tissues is influenced by ECM pattern, where cardiomyocytes seeded on a substrate coated with a uniform layer of FN (Fig 4A) produce a weaker contractile force than those seeded on FN lines 20 μm wide with 5 μm gaps between lines (Fig 4B) [19]. In addition to FN pattern, there are other experimental conditions that can impact the quality of tissue architecture. For example, seeding cardiomyocytes at a low density (Fig 4C) produces sparse tissues that have less developed myofibrils [66].

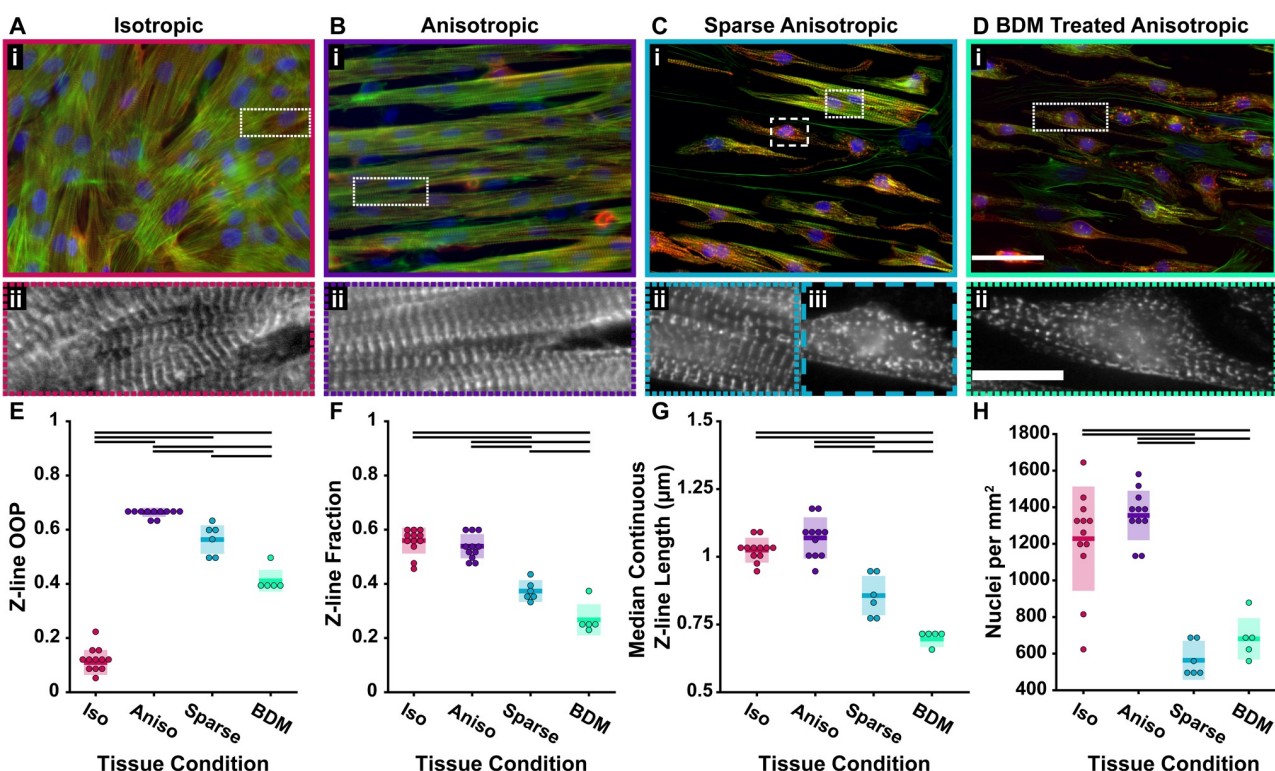

**Fig 4. Comparison of cardiac tissues. A-D**, Cardiac tissue stained for actin (green), $\alpha$-actinin (red), and nuclei (blue) on a uniform layer of FN (**A**), FN in lines (**B**), FN in lines with sparsely seeded cardiomyocytes (**C**), and FN in lines with cardiomyocytes treated with BDM (**D**). **E**, Z-line OOP. **F**, Z-line fractions. **G**, Median continuous z-line lengths. **H**, Nuclei per area. In **E-H** Each dot represents a coverslip. In **E-H**, each dot represents a single coverslip, colored bars represent the mean, and colored boxes represent the standard deviation. Groups were compared using ANOVA with Tukey's test p <0.05 (black bars in **E**, **F**, **G** and **H**). Number of coverslips (cs) for each condition: isotropic cs = 12, anisotropic cs = 11, sparse anisotropic cs = 6, BDM treated cs = 5. Scale bars: **(A-Di)** 50 μm; **(A-D ii)** 15 μm.

Similarly, BDM, which inhibits myosin ATPase, disrupting the actin-myosin interaction, has been qualitatively noted to disrupt z-line registration in cardiac and skeletal muscle cells [28, 31] and shown to cause a decrease in contractile force [67]. Therefore, the z-line architecture was compared between isotropic (Fig 4A), anisotropic (Fig 4B), sparse anisotropic (Fig 4C), and BDM treated anisotropic (Fig 4D) tissues.

As expected, the orientational order of the z-lines was significantly different between isotropic and anisotropic tissues (Fig 4E). While the sparse anisotropic and BDM treated anisotropic tissues tended to follow the orientation of the FN pattern and had higher orientational order than isotropic tissues (Fig 4E), the tissues contained malformed myofibrils and therefore had a lower orientational order than anisotropic tissues. However, the value of the orientational order parameter by itself does not indicate the degree of malformation of tissues or the impact on z-line continuity. In contrast, the z-line fraction was significantly lower in the sparse anisotropic and BDM treated anisotropic tissues than in the isotropic and anisotropic tissues (Fig 4F), which was expected because of off-target $\alpha$-actinin staining along immature stress fibers and cell boundaries [2] (Fig 4Ciii and 4Dii). The z-line fraction was sensitive enough to be able distinguish between anisotropic tissues and either sparse anisotropic or BDM treated anisotropic tissues with as few as 2-3 coverslips of each tissue. Additionally, the median continuous z-line length was significantly lower in both the sparse anisotropic and BDM treated anisotropic tissues than the isotropic and anisotropic (Fig 4G, S4 Fig). While the z-line architecture of the single cells was expertly classified as good, intermediate, or bad, it was impractical to

classify individual cells within whole tissues. Therefore, the z-line fraction of the anisotropic and isotropic tissues falling between the z-line fraction of good and intermediate single cells (Fig 3C) indicated that these tissues contain cells with both good and intermediate z-line architecture. Similarly, the median continuous z-line length for the anisotropic and isotropic tissues was lower than the median continuous z-line length for single cells with good z-line architecture and closer to that of cells with intermediate z-line architecture (Fig 3D).

Both sparse seeding and treatment with BDM produced tissues with poor z-line architecture, which was quantified by the z-line fraction and median continuous z-line length. However, sparse anisotropic tissue contained regions that appeared more developed (Fig 4Cii) and looked similar to well-formed anisotropic tissues (Fig 4Bii), as well as regions with z-line architecture that appeared similar to tissues treated with BDM (Fig 4Ciii and 4Dii). Although the sparse anisotropic and BDM treated anisotropic tissues contained a similar number of cells in a given field of view, as indicated by the number of nuclei (Fig 4H), both the z-line fraction and median continuous z-line length were significantly higher in the sparse anisotropic than the BDM treated anisotropic (Fig 4F and 4G). These results indicate that treating tissues with BDM produced tissue with more disrupted z-line architecture than was produced by seeding at a low density.

## Discussion

In this work, we developed `ZlineDetection`, an image analysis protocol to characterize z-line architecture in $\alpha$-actinin stained striated muscle tissues. Using improved z-line extraction and a biologically motivated approach to segmenting off-target staining, we were able to automatically and accurately isolate z-lines. We also designed and implemented an algorithm to measure the length of continuous z-lines, which is physically related to z-line registration [32, 34, 41, 68, 69]. Improved z-line extraction coupled with a variety of metrics allowed `ZlineDetection` to capture differences in z-line architecture among standard engineered cardiac tissues (Fig 4).

A major achievement of this work was automating z-line isolation, making analysis of z-line architecture in single cells, engineered tissues, and tissue sections [35] possible without the need for experts to trace z-lines [32] or manually segment off-target staining [5, 8, 18, 30, 35]. Segmentation guided by local actin orientation eliminates pixels that correspond to off-target staining, where in manual segmentation it is only possible to eliminate large regions of off-target staining. Consequently, subsequent analysis of the orientational order of z-lines in anisotropic tissues (Fig 4E) resulted in a higher OOP than previously reported when segmentation was done manually [18]. This quantitatively demonstrates the advantages of the `ZlineDetection` algorithm even if only the previously explored metrics are used. Although there are many different approaches to segmentation of biological images [70, 71], in addition to not requiring user training, the actin guided segmentation also provides additional information about the amount of off-target staining, which occurs along immature stress fibers and cell boundaries [2]. Accordingly, the z-line fraction captured differences in the architecture quality between expertly classified single cells (Fig 3C). Furthermore, the z-line fraction was able to successfully distinguish between malformed and well-formed tissues, as we showed that malformed tissues contained more off-target staining than well-formed anisotropic and isotropic tissues (Fig 4G). Without the use of actin guided segmentation, the orientational order parameter of $\alpha$-actinin combined organization of z-lines and the amount of off-target staining, while the z-line fraction and z-line OOP quantified both aspects of architecture separately. Decoupling these two metrics is essential, as although an expert would consider both the isotropic (Fig 4A) and anisotropic (Fig 4B) tissues to be well-formed, the

$\alpha$-actinin OOP alone would indicate that the isotropic tissues were both disorganized and low quality, rather than only disorganized.

In the absence of the actin co-stain, $\alpha$-actinin pixels can no longer be classified as z-lines or off-target staining, and ZlineDetection can no longer decouple these two metrics and instead reports the simple $\alpha$-actinin OOP that has been used previously [5, 18, 30, 32]. Existing algorithms that quantify sarcomere architecture from only a z-line stain, isolate z-lines from off-target staining by using signal processing to identify double wavelets in an image [72], quantify local, micron-scale organization [73], or manually remove off-target staining [5, 18, 30, 32]. These previous works have used a range of metrics to then classify tissue or cell architecture, many of which describe similar properties to those provided by ZlineDetection. For example, the local sarcomere organization algorithm developed by Sutcliffe et al. [73] scored isolated cells without actin co-staining (Figs 4 and 6d in Sutcliffe et al. [73]) with a sarcomere organization index, which ranges from 0-2. The sarcomere organization index [73] was ~0.1 for a primary cell they qualitatively classified as "disorganized", but ~0.4 for both a "well-organized" primary cell and a reprogrammed cardiomyocyte, even though the latter had disorganized myofibrils and some $\alpha$-actinin punctate patterns indicative of premyofibrils. By contrast, the $\alpha$-actinin OOP, which measures organization globally, was 0.2, 0.52, 0.17 for the three cell types, respectively. As such, the cell-tissue scale OOP and the sarcomere scale organization index [73] provide qualitatively different measurements, but the key difference is in the scale at which the measurement is happening. Because tissue level organization influences the strength of tissue contraction [19], we believe a global metric is essential. Nevertheless, the main advantage of using the actin co-stain to isolate z-lines is the ability to quantify the amount of off-target staining based on the additional biological information. This can be especially useful in analyzing noisy stem cell derived cardiomyocytes. For example, a square stem cell derived cardiomyocyte shown in a previous publication [74] was easily analyzed with ZlineDetection (z-line OOP = 0.17, z-line fraction = 0.48, median continuous z-line length = 0.87 μm). While not an interesting metric in primary cardiomyocytes, ZlineDetection also reports the distance between z-lines (i.e. sarcomere length), which is often used to quantify cardiomyocyte maturity [75, 76] and disease state [77].

In addition to the orientational order, the relative spatial location of z-lines in neighboring myofibrils (e.g. continuity or registration of z-lines) has been used to evaluate striated myocytes [34, 41, 68, 69]. In this work, we created and automated an algorithm to measure continuous z-lines by grouping z-line pixels based on relative orientation and location. Consistent with the observation that BDM disrupts the formation of z-lines [28, 31], the median continuous z-line length was significantly lower in BDM treated anisotropic tissues than the malformed tissues that were created by seeding cardiomyocytes at a lower density (Fig 4G). This demonstrates one of the utilities of ZlineDetection is its ability to evaluate the quality of cardiac tissues or other striated muscle without introducing user bias. The median continuous z-line length was not significantly different between isotropic and anisotropic tissues, which suggests that z-line continuity is not impacted by FN pattern. However, because isotropic tissues are weaker than expected based on OOP alone [19], it is worth investigating if differences in z-line registration [20, 39, 40, 42] rather than continuity could be accounting for the difference in contractile strength. Further, although the orientational order and number of z-lines accounts for the low stresses produced by single cells at aspect ratio 1:1 and 14:1 [32](Fig 3F), z-line registration might account for the peak in contractile strength at the aspect ratio ~7:1 compared to other aspect ratios which have similar orientational order and number of z-lines. The suggestion that z-line registration influences contractile strength only in the presence of high orientational order aligns with the liquid crystal view that smectic order (i.e. registration) has no meaning in the absence of high nematic order (i.e. high orientational order).

While `ZlineDetection` is a significant step forward in automating sarcomere architecture, as with any analysis method, it is not without limitations. Although using local actin orientation to classify α-actinin staining as off-target effectively captured differences between well-formed and malformed tissues as well as more accurately isolated z-lines, actin guided segmentation falters in cases where tissues are not organized in a pure myocyte monolayer as fibroblasts can sit on top of or under myocytes (Fig 1D). Therefore, future work involves adapting our analysis protocol to confocal z-stacks and eliminating the actin of non-myocytes. Additionally, the image analysis pipeline and corresponding parameters were optimized for a resolution ~6 pixels/μm. However, if the images are of particularly bad quality (i.e. dominated by background fluorescence) the suggested parameters might need to be adjusted, but we caution users against using images of poor quality (S5 Fig) as information is invariably lost. Another consideration in choosing parameters is that the continuous z-line lengths in particular are sensitive to the amount and duration of smoothing in the anisotropic diffusion filtering step. Although there has been work on calculating the parameters for anisotropic diffusion filtering based on the statistics of the images [78–80], there is not an emphasis on preserving edges and continuity of line segments, which is how we selected our parameters (S2 Fig). Therefore, we made `ZlineDetection` open source and adaptable to improvements and advancements in image analysis and quantitative parameter selection. It is worth noting that `ZlineDetection` can be used with other stains or methods of visualizing cardiac striations. For example, it is possible to analyze videos previously published by Sharma et al. [81] of unfixed cells that express florescent proteins. Such an analysis in the absence of a co-stain would result in OOP of the expressed protein labeled structures and the striation lengths as a function of time (S6 Fig). While the median is a useful metric to summarize a skewed distribution and the median continuous z-line length was significantly different between malformed and well-formed tissues (Fig 4G), it is possible that the relative location and distribution of continuous z-lines are important predictors of stress generation. Therefore, future work includes creating a method to measure registration and examining its relationship with z-line continuity and contractile function. Further, the ability to measure and quantify registration could be impactful in other fields, because registration of cellular structures may be important in different cell types [40, 42].

## Conclusion

Our image analysis protocol and implementation can be used as a tool to quantitatively compare z-line architecture in single cells and tissues under different conditions and between labs. This would increase the scientific rigor in the field by eliminating qualitative and/or manual analysis that introduces lab/user specific bias. Indeed, one of the key achievements of `ZlineDetection` is the ability to segment off-target staining automatically. An additional advantage of the biologically motivated segmentation approach was that the amount of off-target staining can be used to compare quality of tissue formation. A second key achievement was identifying experts' criteria for evaluating the quality of striated tissues, as the z-line orientational order, intact z-line fraction, and the relative spatial alignment of z-lines. A third key achievement of this capability was elucidating the mechanism by which cardiomyocytes with an elongated aspect ratio become inefficient at producing force. Having the ability to measure continuous z-lines can pave the way to predicting force measurements as it relates to z-line architecture in cardiac tissues through the use of experimental and mathematical modeling approaches. Finally, in the future these computational methods can be used as a quality control to analyze z-line architecture in stem cell derived myocytes, engineered tissues, diseased tissues, and tissues subjected to injury and treatment with pharmacological agents.

## Supporting information

**S1 Table. Description of parameters used by `ZlineDetection`.** Column 1 is the stage of analysis at which a parameter is used, followed by its description in column 2. The third column is the value of the parameters used for analysis of the images, which had the resolution ~6 pixels/μm.
(PDF)

**S1 Fig. Image analysis workflow.** Squared boxed text indicates an image analysis step in `ZlineDetection`. Rounded boxed text indicates an image or matrix, where binary skeletons are shaded gray, matrices containing information about $\alpha$-actinin stained images are shaded red, and matrices containing information about actin stained images are shaded green. Purple circles indicate matrix multiplication. On a computer with 32 GB of RAM, `ZlineDetection` took ~30 s to analyze a 1024 x 1344 image.
(PDF)

**S2 Fig. Selection of diffusion filter parameters. A**, Sections of anisotropic, isotropic, and sparse anisotropic tissues with manually traced z-lines, which was done three times. **B**, Similarity between the three different manual traces of z-lines. **C**, Average similarity for each set of diffusion filtering parameters. **D**, Average similarity for a more refined range of diffusion filtering parameters. **E**, Error for refined range of diffusion filtering parameters (Eq 1).
(PDF)

**S3 Fig. Z-line tissue metrics by aspect ratio.** For the good cells of each aspect ratio, the mean and standard deviation are shown for the median continuous z-line length (**A**), mean continuous z-line length (**B**), total continuous z-line length (**C**), z-line OOP (**D**), and the estimated force along the axis perpendicular to principle axis (**E**). **F**, Normalized distribution of continuous z-line lengths for the representative cells in Fig 3. Groups were compared using ANOVA with Tukey's test $p < 0.05$ (black horizontal bars above data).
(PDF)

**S4 Fig. Continuous z-lines plotted on tissue segments.** Sections of cardiac tissue shown in Fig 4A–4D stained for actin (green) and $\alpha$-actinin (red) on a uniform layer of FN (**Ai**), FN in lines (**Bi**), FN in lines with sparsely seeded cardiomyocytes (**Ci**), and FN in lines with cardiomyocytes treated with BDM (**Di**) and their corresponding continuous z-lines (**A-Dii**). Scale bar: 15 μm.
(PDF)

**S5 Fig. Examples of good and poor imaging. A**, $\alpha$-actinin stained image of poor imaging quality. **B**, Example of good imaging quality. For both **A** and **B**, the background (**i**), foreground (**ii**), and distribution of intensities (**iii**) are shown. Scale bars: (**A-B i**) 20 μm; (**A-B ii-iii**) 10 μm.
(PDF)

**S6 Fig. Analysis of beating cardiomyocyte.** Results of analyzing titin-GFP sarcomere reporter human induced pluripotent stem cell-derived cardiomyocyte published by Sharma et al. [81]. (**A**) Orientational order parameter and (**B**) median continuous z-line length in pixels as a function of frame number. As expected, the OOP was relatively constant throughout the contraction, while the median continuous z-line length varied due to non-synchronous contractions of neighboring myofibrils.
(PDF)

## Acknowledgments

We thank Professor Samuel Safran and Ohad Cohen (Weizmann Institute) for their discussions on sarcomeric continuity and registration. We also thank Dr. Rabi Tawil (University of Rochester Medical Center) for providing the original control myoblasts, and Emil Martin Lundqvist (University of California Irvine) for his discussions on imaging.

## Author Contributions

**Conceptualization:** Tessa Altair Morris, Anna Grosberg.

**Data curation:** Tessa Altair Morris, Jasmine Naik, Kirby Sinclair Fibben, Xiangduo Kong, Tohru Kiyono, Kyoko Yokomori.

**Formal analysis:** Tessa Altair Morris, Anna Grosberg.

**Funding acquisition:** Tohru Kiyono, Kyoko Yokomori, Anna Grosberg.

**Investigation:** Tessa Altair Morris, Anna Grosberg.

**Methodology:** Tessa Altair Morris, Anna Grosberg.

**Project administration:** Anna Grosberg.

**Resources:** Anna Grosberg.

**Software:** Tessa Altair Morris.

**Supervision:** Anna Grosberg.

**Validation:** Tessa Altair Morris, Anna Grosberg.

**Visualization:** Tessa Altair Morris, Anna Grosberg.

**Writing – original draft:** Tessa Altair Morris, Anna Grosberg.

**Writing – review & editing:** Tessa Altair Morris, Anna Grosberg.

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
