## [Decision Letter · Decision Letter 0]

5 Dec 2019

Dear Dr Grosberg,

Thank you very much for submitting your manuscript 'Striated myocyte structural integrity: automated analysis of sarcomeric z-discs' for review by PLOS Computational Biology. Your manuscript has been fully evaluated by the PLOS Computational Biology editorial team and in this case also by independent peer reviewers. The reviewers appreciated the attention to an important problem and the utility of this method, but raised some substantial concerns about the manuscript as it currently stands. In particular, Reviewer 2 noted important comparisons to other methods for sarcomere analysis, imaging, or cell types. Addressing these comments may involve but does not necessarily require additional experiments. The computer code could be made available at a repository with a private link for review only, and the GitHub repository could be verified if the manuscript is accepted. While your manuscript cannot be accepted in its present form, we are willing to consider a revised version in which the issues raised by the reviewers have been adequately addressed. We cannot, of course, promise publication at that time.

Sincerely,

Jeffrey J. Saucerman

Associate Editor

PLOS Computational Biology

Jason Haugh

Deputy Editor

PLOS Computational Biology

[LINK]

Reviewer's Responses to Questions

**Comments to the Authors:**

Reviewer #1: The authors present a set of image analysis tools that enable quantification of z-line architecture in contracting myocytes, and use these to elucidated the mechanisms by which highly elongated cardiomyocytes lose efficiency in force production. Their code ZlineDetection evaluates sarcomere architecture in a way that mimics expert imaging analysis, and appears to be a general tool that enables this analysis to occur objectively. The methods are straightforward and robust. The examples shown demonstrate clear utility for the methods. The reviewer believes that the software meets a clear need in the field.

The only disappointment with the paper was that the GitHub link did not work; the reviewer would have liked to have tried the software.

Reviewer #2: The manuscript entitled ‘Striated myocyte structural integrity: automated analysis of sarcomeric z-discs’ describes a MatLab algorithm that can be used to assess the order of sarcomeres in striated muscle cells. This is an automated technique that uses z-disc and actin co-staining to reduce z-disc characterization error. The technique provides a set of parameters to determine characteristics of z-disc architecture and differences generated by experimental treatments. The technique will undoubtedly be of interest to the field, although not the only algorithm used to make these types of measures. Nevertheless, useful alongside existing technologies.

Comments:

How does this algorithm compare to that of for instance Sutcliffe et.al. (PMID: 29352247) and others? What are the pros and cons of using other techniques versus the one presented in this manuscript? Could one make a direct comparison between the techniques to highlight what this current algorithm achieves over previous methods?

The technique has been tested on cultured myocytes as well as primary myocytes, has this at all been tested in unstructured iPSC derived cardiomyocytes? Especially those that have not been cultured on a patterned surface. These would be cells that have a far higher variance in z-line structure. This could be of interest to a broad audience, due to the emergence of many models of disease in iPSC derived cardiomyocytes that display sarcomere malformations.

Is the algorithm reliant on the co-stain with actin or can it be run with just a z-disc label? Maybe this has already been stated somewhere. This could be a particular hurdle when looking at relatively unstructured iPSC derived cardiomyocytes.

Can the system be used on unfixed cells that have endogenous tags, or express fluorescent proteins for instance PMID 29364522 and 30956114.

When referencing z-line length, is there an explanation why the z-line length in figure 4 Iso and Aniso are so much lower than in figure 3i and 3ii? Is this borne out in the literature? Relatively speaking the error of the z-line measure in figure 3 has a far higher error, which is maybe surprising given that these are from patterned cells

Presumably the algorithm doesn’t output sarcomere length for pairs of these z-lines in the resting stained cells (see PMID 30700234)? It could be of interest in many models of disease, drug application, and development to see if there is a correlate between resting sarcomere length and sarcomeric order.

Minor:

What is the computing power needed to run this script. It would be useful to know how long a set of images take to run through the pipeline and if this can be done through a computing cluster if need be.

Figure 3F the Y-axis label has been lost. In general the main text figures are very low resolution, making it hard to see the images clearly. The images in the supplement are of far higher resolution.

The Discussion / Conclusion section states that this system can aid in modelling force outputs, I think this statement may need to be reconsidered. Would this technique for instance predict changes in force, if the change in force were not accompanied by a change in z-line architecture? Such as could be hypothesized in hypertrophic or dilated cardiomyopathies.

Figure 4 bars for mean are not visible on my version, maybe this is due to a compression issue on submission. Please check.

Figure 3 A you show the markups of the z-lines identified in the patterned cells, it would be nice to see how reliable this markup is in Figure 4 panels Aii, Bii, Cii and iii and Dii.

Z-line is sometimes capitalized in figures and sometimes lower case z-line.

**Have all data underlying the figures and results presented in the manuscript been provided?**

Reviewer #1: Yes

Reviewer #2: Yes

PLOS authors have the option to publish the peer review history of their article (what does this mean?). If published, this will include your full peer review and any attached files.

Reviewer #1: No

Reviewer #2: No

---

## [Decision Letter · Decision Letter 1]

23 Jan 2020

Dear Dr. Grosberg,

We are pleased to inform you that your manuscript 'Striated myocyte structural integrity: automated analysis of sarcomeric z-discs' has been provisionally accepted for publication in PLOS Computational Biology.

Before your manuscript can be formally accepted you will need to complete some formatting changes, which you will receive in a follow up email. A member of our team will be in touch within two working days with a set of requests.

Best regards,

Jeffrey J. Saucerman

Associate Editor

PLOS Computational Biology

Jason Haugh

Deputy Editor

PLOS Computational Biology

Reviewer's Responses to Questions

**Comments to the Authors:**

Reviewer #1: The software was very simple to use-- thanks for uploading! All concerns have been addressed.

Reviewer #2: The revised manuscript has comprehensively answered my questions. I have no further comments.

**Have all data underlying the figures and results presented in the manuscript been provided?**

Reviewer #1: Yes

Reviewer #2: Yes

PLOS authors have the option to publish the peer review history of their article (what does this mean?). If published, this will include your full peer review and any attached files.

Reviewer #1: No

Reviewer #2: No

---

## [Editor Report · Acceptance letter]

26 Feb 2020

PCOMPBIOL-D-19-01798R1 

Striated myocyte structural integrity: automated analysis of sarcomeric z-discs

Dear Dr Grosberg,

I am pleased to inform you that your manuscript has been formally accepted for publication in PLOS Computational Biology. Your manuscript is now with our production department and you will be notified of the publication date in due course.

With kind regards,

Sarah Hammond
